

# Evaluation of a Unique Approach to High-Resolution Climate Modelling using the Model for Prediction Across Scales (MPAS) version 5.1

Allison C. Michaelis[1], Gary M. Lackmann[2], Walter A. Robinson[2]

[1]Center for Western Weather and Water Extremes, Scripps Institution of Oceanography, La Jolla, CA 92037, USA
[2]Department of Marine, Earth, and Atmospheric Sciences, North Carolina State University, Raleigh, NC 27695, USA

*Correspondence to*: Allison Michaelis (acamras@ncsu.edu)

**Abstract.** We present multi-seasonal simulations representative of present-day and future thermodynamic environments using the global Model for Prediction Across Scales-Atmosphere (MPAS) version 5.1 with high resolution (15 km)
throughout the Northern Hemisphere. We select ten simulation years with varying phases of El Niño-Southern Oscillation (ENSO) and integrate each for 14.5 months. We use analysed sea surface temperature (SST) patterns for present-day simulations. For the future climate simulations, we alter present-day SSTs by applying monthly-averaged temperature changes derived from a 20-member ensemble of Coupled Model Intercomparison Project Phase 5 (CMIP5) general circulation models (GCMs) following the Representative Concentration Pathway (RCP) 8.5 emissions scenario. Daily sea ice
fields, obtained from the monthly-averaged CMIP5 ensemble mean sea ice, are used for present-day and future simulations. The present-day simulations provide a reasonable reproduction of large-scale atmospheric features in the Northern Hemisphere such as the wintertime midlatitude storm tracks, upper-tropospheric jets, and maritime sea-level pressure features as well as annual precipitation patterns across the tropics. The simulations also adequately represent tropical cyclone (TC) characteristics such as strength, spatial distribution, and seasonal cycles for most of Northern Hemispheric basins.
These results demonstrate the applicability of these model simulations for future studies examining climate change effects on various Northern Hemispheric phenomena, and, more generally, the utility of MPAS for studying climate change at spatial scales generally unachievable in GCMs.

**Plain Language Summary.** We expect that high-impact weather events will change in a warmer climate. Computational
constraints limit global climate models to resolutions that are too coarse to fully capture many societally significant weather events, such as tropical cyclones and flooding rains in middle-latitude low-pressure systems. While these global models often provide reliable projections of changes in mean temperatures and global circulation patterns, they cannot tell us how intense, high-impact events may be altered in response to climate change. Here, we present a novel set of atmospheric simulations designed to address changes in high-impact weather events. The model covers the globe, but has higher
resolution in the Northern Hemisphere. We simulate ten years sampling a range of tropical climate conditions, as represented by observed ocean surface temperatures, and we carry out simulations for current and projected late 21st-century climate



conditions. The future runs include increased heat-trapping gases, higher ocean temperatures, and reduced polar sea ice as projected by conventional global climate models. Our model reasonably replicates present-day climate features, such as large-scale precipitation patterns, mid-latitude cyclone activity, the jet stream, and tropical cyclone characteristics. It reproduces features of climate change that are expected from global climate models, but it also captures smaller scale, high-

impact weather events. We anticipate that these simulations will have great value in understanding changes in high-impact weather events.

## 1 Introduction

We present a novel approach to high-resolution climate modelling with the intent of examining the effects of climate change on high-impact Northern Hemispheric weather phenomena. It is nearly certain that rising global greenhouse gas

concentrations over the next century will result in significant changes to the Earth's climate system (IPCC, 2014). Further understanding of how climate change will affect global and regional weather is essential to informing the scientific community, stakeholders, and policymakers on what actions should be taken to prepare for the future. Here, we present a unique set of high-resolution, multi-seasonal, global, atmosphere-only simulations conducted with the Model for Prediction Across Scales (MPAS; Skamarock et al., 2012) in present and future thermodynamic environments for this purpose: to study

climate change effects on Northern Hemispheric weather phenomena, including extreme events. Through its variable resolution grids, MPAS offers the possibility of investigating local weather phenomena at high resolution in the context of a global model, while avoiding the prohibitive demands on computational resources entailed by running a model globally at high resolution. To our knowledge, however, climate change experiments at long integration times are a novel application of MPAS. Therefore, in order to demonstrate their utility for addressing climate change effects on high-impact weather events,

it is necessary to evaluate how large-scale circulations and responses to warming are represented in such simulations, thus defining the objective of this paper.

      With simulations spanning several centuries, multiple ensemble members, and the inclusion of atmosphere-ocean coupling, the latest generation of general circulation models (GCMs) from the Coupled Model Intercomparison Project Phase 5 (CMIP5) are common tools for determining the effects of climate change. Due to current computational limitations,

however, the grid spacing of these simulations is largely restricted to ~1º (~100 km) or greater. While this coarse resolution is suitable for representing large-scale atmospheric features such as the polar amplification of global warming and teleconnections, it is insufficient for resolving weather extremes, especially those associated with smaller-scale systems such as tropical cyclones, mesoscale features within extratropical cyclones, and convective storms (e.g., Mizielinski et al., 2014 and references therein; Small et al., 2014; Prein et al., 2015; Haarsma et al., 2016; Roberts et al., 2018). These smaller-scale

systems often result in significant socioeconomic impacts; therefore, in order to fully ascertain the societal impacts of climate change, it is essential to complement existing GCM simulations with simulations at resolutions sufficiently fine to capture these high-impact phenomena. The ongoing High Resolution Model Intercomparison Project (HighResMIP;

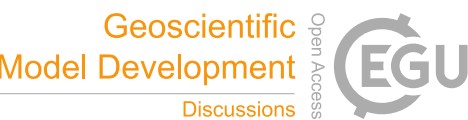



Haarsma et al., 2016) associated with the CMIP6 will also be highly beneficial in understanding how weather extremes respond to climate change.

To date, several model downscaling techniques have been developed for this purpose. For example, regional downscaling (e.g., Wang et al., 2004; Giorgi et al., 2009) computationally allows for finer grid spacings by employing a smaller domain, thus circumventing the resolution deficiency of traditional GCMs. Using a regional domain, however, presents the issue of how to specify lateral boundary conditions, and two-way interactions with larger-scales cannot be fully incorporated (Small et al., 2014). Global models eliminate the constraints of lateral boundaries, but are expensive to run for long periods at high resolutions. Incorporating nests within a global domain, or using mesh refinement grids, however, can help alleviate this expense.

Another useful method for assessing climate change effects is the "pseudo-global warming" (PGW) method, initially called "surrogate global warming" (e.g., Schär et al., 1996; Frei et al., 1998; Kimura and Kitoh, 2007; Hara et al., 2008; Rasmussen et al., 2011; Mallard et al., 2013; Lackmann, 2013, 2015; Trapp and Hoogewind, 2016). In PGW experiments, high-resolution control simulations are conducted, typically replicating an observed weather event. The high-resolution initial and boundary conditions are then modified with "delta" fields derived from GCMs, and the event is re-simulated, allowing assessment of changes in the characteristics of the event as a function of larger-scale environmental change. An important advantage of the PGW method is that realistic, high-resolution synoptic-scale and mesoscale settings are guaranteed. This method is consistent with the "storyline" approach described by Shepherd (2016), Hazeleger et al. (2015), and Trenberth et al. (2015). A limitation of PGW case studies is the inability to study the frequency of occurrence of such events. To alleviate this limitation, some investigators have conducted long-duration regional PGW simulations (e.g., Ban et al., 2014; Willison et al., 2015; Liu et al., 2017), which allow for analysis of statistical changes extending beyond the case-study of a single event. All regional PGW experiments, however, are limited by the need to impose lateral boundary conditions, which reduces the dynamical freedom of the simulations.

Given recent advances in computational power and data storage, several modelling groups have performed long-term high-resolution global simulations, both with atmosphere-only and with coupled atmosphere-ocean configurations (e.g., Small et al., 2014; Komada et al., 2015; Murakami et al., 2015; Roberts et al., 2015 and references therein; Haarsma et al., 2016; Yamada et al., 2017). Models that include coupling between the atmosphere and ocean have the advantage of two-way communication, allowing the possibility of realistic atmosphere-ocean interactions. At long integration times, however, climatologies of coupled models have been known to suffer from biases due to the drift in sea surface temperatures (SSTs), which can negatively affect regional climate projections (e.g., He and Soden, 2016). Previous studies have determined that resolution is an important factor for a more accurate representation of synoptic and mesoscale phenomena in the atmosphere (Willison et al., 2013; Small et al., 2014; Prein et al., 2015), and thus should be maximized whenever possible.

The present global simulations use a 15 km grid in the Northern Hemisphere, relaxing to a 60 km grid in the Southern Hemisphere to reduce computational expense (Fig. 1). These simulations are conducted with MPAS as individual time-slice runs, selected to span a range of El Niño-Southern Oscillation (ENSO) states. We use high-resolution SST



analyses that capture oceanic eddies and fronts, which have been shown to exert an important influence on atmospheric variability (e.g., Kirtman et al., 2012; Siqueira and Kirtman, 2016; Ma et al., 2017; Parfitt et al., 2017). Our novel modelling approach aims to eliminate several limitations from the previously discussed methods. For one, using a global model circumvents issues related to lateral boundary conditions, thus improving upon limited-area simulations; here, higher-

resolution in the Northern Hemisphere was obtained at the expense of reduced resolution in the Southern Hemisphere. Our future simulations are similar to PGW in the treatment of SST in that we apply a GCM-based delta to analysed SST fields; this incorporation of high-resolution SSTs precludes issues of the type noted by He and Soden (2016), and represents a potential improvement to coupled atmosphere-ocean model configurations. The atmospheric resolution of 15 km is sufficiently high to represent strong tropical and extratropical cyclones as well as flooding rainfall, which is a considerable

improvement compared to GCM simulations. Furthermore, the sample size is sufficient to allow statistical comparisons of features such as Northern Hemisphere storm tracks and tropical cyclone activity, thus improving upon the traditional PGW case study approach.

We present our simulations with the intention of providing an additional realization of a complex system in order to improve our understanding of potential climate change effects on Northern Hemispheric high-impact weather phenomena.

For such a modelling system to be useful for this purpose it is necessary that:

- It reproduces the present-day climate and global circulation of the atmosphere.
- It demonstrates the benefits of enhanced resolution in simulating high-impact weather phenomena.
- It provides simulations of a future climate consistent with expectations derived from GCMs.

Thus, in the present paper, following a discussion of the model and how our simulations were conducted (section 2), we

offer analyses of its representation of the present-day climate (section 3), and its simulation of Northern Hemisphere tropical cyclones and their climatology within the present-day climate (section 4). We focus on tropical cyclones as an exemplar of high-impact weather phenomena that are challenging to represent accurately in models, and for which successful simulation demands high resolution (e.g., Davis, 2018). In section 5 we examine the model representation of climate change in response to global warming boundary conditions. Last, section 6 presents a summary of our findings and discusses future applications

of our simulations for investigating how high-impact weather may change in a warmer climate.

## 2 Models, Experiments, and Performance

### 2.1 Model Configuration

We conduct our simulations using the atmospheric component of MPAS, version 5.1 (Skamarock et al., 2012). MPAS is a non-hydrostatic global, atmosphere-only model that uses unstructured Voronoi meshes (Du et al., 1999) to create variable-

resolution grids. This grid structure permits localized areas of high resolution to transition gradually to lower resolutions, thus alleviating the boundary issues associated with sharp transitions between domains in traditional nesting approaches (Park et al., 2014). The focus of the simulations presented here is on Northern Hemispheric phenomena; therefore, we use a



variable-resolution mesh with 15 km grid spacing over the Northern Hemisphere, expanding out to 60 km in the Southern Hemisphere (Fig. 1).

The MPAS atmospheric physics suite includes a subset of schemes adapted from versions of the Weather Research and Forecasting (WRF) model (Skamarock et al., 2008). Our simulations employ the following physics parameterizations:

WRF Single-Moment 6 class (WSM6; as in WRF 3.8.1) microphysics, Yonsei University (YSU; as in WRF 3.8.1) representation of the planetary boundary layer, Tiedtke (as in WRF 3.3.1) sub-grid scale convective parameterization, Community Atmosphere Model (CAM; as in WRF 3.3.1) shortwave and longwave radiation, and the Noah land surface model (as in WRF 3.3.1) for surface processes. We selected the Tiedtke convective parameterization scheme because it includes convective momentum transport (CMT), which has been shown to be important for reducing model biases in

surface winds and tropical cyclone (TC) intensity (Zhang and McFarlane, 1995; Han and Pan, 2006; Hogan and Pauley, 2007; Richter and Rasch, 2008). CMT also improves the representation of features such as the Intertropical Convergence Zone (Zhang and Wang, 2006; Kim et al., 2008). We completed a series of preliminary tests using a quasi-uniform 60 km mesh to further refine our physics choices (not shown).

## 2.2 Present-Day and Future Climate Simulations

We selected ten simulation years with varying phases of ENSO based on the Multivariate ENSO Index (MEI) and the Oceanic Niño Index (ONI) over the TC season (Table 1). These years were also chosen to sample a range of TC activity in the North Atlantic, Eastern North Pacific, and Western North Pacific basins. Each simulation is integrated for 14.5 months, from 1 March of the first year through 14 May of the following year, with the first month discarded as spin-up; output is recorded every 6 h.

We used the ECMWF Interim Reanalysis (ERA-I; Dee et al., 2011) with a spectral T255 resolution (~0.7º horizontal grid spacing) for present-day initial conditions. SST and sea ice fields are updated daily throughout the simulations. The configuration of these surface fields is discussed further in Section 2.3. For the future climate simulations, we modify the ERA-I initial and lower boundary conditions by adding monthly-averaged temperature changes derived from a 20-member ensemble of CMIP5 GCMs (Table 2). These temperature changes are calculated by subtracting the 1980–1999

average temperature from the 2080–2099 average temperature following the Intergovernmental Panel on Climate Change (IPCC) Fifth Assessment Report (AR5) Representative Concentration Pathway (RCP) 8.5 emissions scenario, interpolated to the ERA-I grid, and added to the existing temperature data at all atmospheric pressure and soil levels. Geopotential height and specific humidity are adjusted by the model based on the imposed temperature changes; relative humidity is held constant at the initial time. We set carbon dioxide ($CO_2$) concentrations in the future climate simulations to 936 ppm, the

level projected by the RCP8.5 emissions scenario for 2100 (Meinshausen et al., 2011). Present-day $CO_2$ concentrations are based on analysed values set according to the year.

Rather than running the simulations in chronological order, the simulation years are sorted from the strongest La Niña year (i.e., the year with the smallest MEI and most negative ONI; Table 1) to the strongest El Niño year (i.e., the year





with the largest MEI and most positive ONI; Table 1). This design aims to minimize model spin-up in response to changes in SST. With the present-day and future initial conditions set, we conduct full simulations for a neutral ENSO year (e.g., 2013) in each thermodynamic environment. These simulations are used as spin-up and are, therefore, excluded from our analysis. While we took this precaution to allow the model atmosphere to come into equilibrium with the imposed warming and

adjusted $CO_2$ for the future climate experiment, we repeated this process for the present-day simulation to maintain consistency. We then used the output from 1 March, towards the end of the initial spin-up simulation, to initialise the first simulation year. This method continues for both the present-day and future experiments by using the output from the latter part of one simulation (e.g., 1 March) to initialise the next (Fig. 2). Applying this unique "daisy-chain" technique avoids the need for excessive spin-up times for each year; instead, we discard only the output from the first month, which allows any

discontinuities arising from the change in SST to equilibrate.

### 2.3 Lower Boundary Conditions

As with the atmospheric initial conditions, the SST fields used in the simulations are taken primarily from the ERA-I. The SSTs in the ERA-I have, however, been derived from several different datasets over the years (Dee et al., 2011). For reanalysis times after February 2009, ERA-I surface fields originate from the Operational Sea Surface Temperature and Sea-

Ice Analysis (OSTIA; Donlon et al., 2012). To maintain consistency between all simulations, the OSTIA SST, interpolated from its native 0.05º horizontal grid spacing to the ERA-I grid, is used for simulation years prior to 2009. Therefore, we effectively use OSTIA SST for all simulations. For present-day soil temperature and moisture, we use the ERA-I fields.

The SSTs for the future climate simulations are altered in the same manner as the initial condition atmospheric and soil temperatures (e.g., Fig. 3a–b). The same technique of adding a GCM delta field onto existing data cannot, however, be

used for sea ice. Instead, similar to Mizielinski et al. (2014), monthly-averaged CMIP5 ensemble mean sea ice fields are temporally interpolated to create daily sea ice fields for both present-day (1980–1999) and future (2080–2099 under the RCP8.5 emissions scenario) time periods. An example of these sea ice fields is shown in Figure 3c–d. We then replaced the analysed sea ice in the ERA-I with these climatological fields for use in all model simulations. While the climatological present-day sea ice does not entirely match the analysed field in the ERA-I (e.g., the sea ice edge is much more diffuse),

handling the sea ice in this manner ensures that it is plausibly represented in the future climate simulations. The presence of an overly diffuse ice edge could result in unrealistically weak lower tropospheric baroclinicity during warm seasons in these locations.

Our technique for simulating a future climate is similar to the PGW approach in the sense that (1) the analysed initial and lower boundary conditions are altered by adding projected temperature changes from GCMs to represent future

thermodynamic conditions, and (2) analysed high-resolution SST fields are used to preserve realistic representation of ocean eddies and SST gradients. High-resolution SST is of demonstrated importance for midlatitude cyclone development and other regional climate changes (e.g., Brayshaw et al., 2011; Booth et al., 2012; Kirtman et al., 2012; He and Soden, 2016; Siqueira and Kirtman, 2016). By using a global model, however, one of the main limitations of PGW, the constraint of the





lateral boundary conditions, is alleviated. The UPSCALE experiments described by Mizielinksi et al. (2014) use a similar time-slice technique for simulating a future climate. By simulating a small ensemble of 26 years, UPSCALE samples a broad range of interannual variability and ENSO states; however, 25 km grid spacing is insufficient for resolving full strength tropical cyclones (Davis, 2018). Therefore, our simulations complement UPSCALE by offering sufficiently high resolution
to better capture the atmospheric mesoscale, specifically tropical cyclones.

### 2.4 Computational Performance

We conducted the MPAS simulations on the National Center for Atmospheric Research (NCAR) supercomputer, Cheyenne (Computational and Information Systems Laboratory, 2017). Cheyenne is a 5.34 petaflop SGI ICE XA Cluster with 145,152 Intel Xeon processor cores and 313 terabytes (TB) of memory. Each 14.5-month simulation was run on 1152 cores and
consumed roughly 92,000 CPU hours, including resources needed for post-processing, leading to a total of ~1.9 million core hours used for these experiments.

We post-process model output to vertically interpolate fields to selected isobaric levels and horizontally interpolate from the native unstructured mesh to a 0.15º x 0.15º latitude-longitude grid. Due to storage constraints, we saved a limited number of variables for the Northern Hemisphere only; however, monthly restart files are archived, enabling replication of a
particular period of time or event as needed. The post-processed output occupies approximately 50 TB of storage space for the output for all 20 simulations, and is currently stored on Cheyenne's High Performance Storage Space (HPSS) and at North Carolina State University.

### 2.5 Assumptions and Limitations

For our future climate simulations, we computed temperature delta values using the mean IPCC AR5 RCP8.5 emissions
pathway. While other plausible scenarios exist, we selected a high emission pathway to maximize the signal of climate change in our simulations. Using the GCM ensemble mean temperature changes to alter our initial and lower boundary conditions diminishes the considerable amount of variability in the temperature changes projected by individual GCMs. Computing an ensemble mean from a set of simulations using temperature change fields from each GCM is, however, unlikely to produce significantly different results (Hill, 2010; Lackmann, 2015; Marciano, 2014).
The adjustment of geopotential height based on the imposed temperature changes for the future climate simulation introduces some degree of imbalance between the model mass and wind fields. In previous studies, we utilized the Digital Filter Initialization (DFI) capability of the WRF model to reduce these imbalances. Since this feature is not available in MPAS, we conducted a full 14.5-month spin-up simulation to allow time for the dynamics of the model atmosphere to restore balance. We also maintain constant relative humidity between the present-day and future simulations in the initial
conditions. While this assumption may be appropriate over ocean basins, it does not necessarily hold true over land areas (e.g., Sherwood and Fu, 2014). With no constraints on the lateral boundaries, however, this constraint is not enforced, as the





relative humidity within our model domain evolves freely through the duration of the simulations; by the end the spin-up period, we expect the distribution of water vapour is fully equilibrated with the simulated future climate.

While our treatment of sea ice in the model allows for a plausible representation of future conditions, we use identical sea ice fields in each member of our present-day simulation set, and similarly for the future set. We therefore

exclude the effects of interannual variability in sea ice. Several studies have highlighted the connection between sea ice variability and atmospheric circulations in the Northern Hemisphere (e.g., Deser et al., 2000; Overland and Wang, 2010); our intention here, however, is to minimize this influence and instead, focus on changes due to altered thermodynamics. Another limitation inherent in our methods is the assumption that future patterns of SST variability will remain similar to what they are today. Nevertheless, we believe the benefits of using high-resolution SST analyses to preserve realistic SST gradients

and alleviate regional biases associated with atmosphere-ocean coupling (e.g., He and Soden, 2016) outweigh this limitation.

Many previous studies have shown that neglect of SST cooling due to cyclone passage results in TCs that are too strong (e.g., Schade and Emanuel, 2009). Use of analysed SST fields in our simulations does not allow for TC-generated cold wakes, which could contribute towards a positive bias in TC intensities and could lead to unrealistic temporal clustering of TCs. The use of convective parameterization, however, particularly the Tiedtke scheme which adjusts momentum, tends

to weaken TCs through momentum adjustment in a warm-core cyclonic structure, an effect opposite to that resulting from the neglect of SST cooling. Ideally, a grid length of 4 km or less would be used to fully capture TC structure and intensity (e.g., Gentry and Lackmann, 2010), but computational expense does not allow this for the Northern Hemisphere region of interest for the simulation durations necessary to obtain statistically meaningful results regarding the impacts of climate change. A benefit of our configuration is that the resolution is sufficiently high to capture nearly the full range of TC

intensity; preliminary testing highlighted the capability of our 15 km grid to replicate realistic TC structures, including spiral rain bands and a defined eye (not shown). We acknowledge that the neglect of sea-surface cooling and the use of parameterized convection are limitations to our approach. These limitations are, however, consistent between present-day and future simulations, allowing any differences found in TC intensity to remain meaningful (Patricola and Wehner, 2018).

We recognize that the methods employed in this study account only for projected changes due to increased

anthropogenic greenhouse gases and therefore, do not represent other external climate forcings. Changes in other aspects of the climate system, such as changes in aerosols, deep soil moisture, and vegetation, are not represented. Despite the limitations discussed, our method alleviates limitations associated with regional PGW and coarse GCMS, and is much more computationally efficient than running high-resolution global models for long integration periods (e.g., centuries); the result is a set of controlled simulations suitable for examining the effects of altered thermodynamics on high-impact weather

events.



## 3 Model Climate: Precipitation and Midlatitude Features

### 3.1 Extratropical Storm Tracks

There are two primary midlatitude storm track regions in the Northern Hemisphere, the North Pacific and the North Atlantic, where baroclinic waves form over regions of enhanced temperature contrast linked to warm western boundary currents off

the east coasts of Asia and North America and propagate eastward through downstream development (Chang et al., 2002). The extratropical cyclones in these regions play an essential role in the Earth's climate system and contribute to everyday weather, including high-impact events. Therefore, it is important that they are well represented in model simulations.

As suggested by Chang and Fu (2003), variance in daily-mean fields can be used as proxies for storm track activity. Here, we use the 24-h variance of daily-mean sea-level pressure (SLP), calculated using Eq. (2) from Chang et al. (2013):

$$SLP\ variance = \overline{[SLP(t + 24h) - SLP(t)]^2}\ , \tag{1}$$

where the overbar indicates the quantity is averaged over time, in this case over the winter season (December–February; DJF) when storm activity in the Northern Hemisphere is maximized (Chang et al., 2002; Brayshaw et al., 2009). Figure 4

shows the wintertime SLP variance for the MPAS simulations compared to the ERA-I; the ERA-I climatology in Figure 4b is computed using only the ten years corresponding to our simulations. The North Pacific and North Atlantic storm track regions are clearly evident in the model simulations; the overall spatial correlation coefficient is greater than 0.98, indicating that general patterns of SLP variance are well reproduced in the MPAS simulations. As evident by the positive biases ~35º N, 165º W and ~40º N, 30º W (Fig. 4c), both storm track regions are shifted equatorward, and the North Pacific storm track

is more zonally oriented in the MPAS simulations. Comparison with 100 random samples of 10-year means from the ERA-I record indicates that these biases, primarily the shift in the North Pacific, likely represent true differences between the MPAS simulations and the real atmosphere. Negative biases in simulated storm activity occur east of Greenland, over Scandinavia, and throughout central North America; these differences, however, fall within the range of observed variability (Fig. 4c).

### 3.2 Northern Hemispheric Jet and Sea-Level Pressure Features

Corresponding to the Northern Hemispheric extratropical storm tracks are the midlatitude jet features, represented by the wintertime average zonal wind speed in Figure 5. As in Figure 4, the ERA-I climatology in Figure 5b includes only the ten simulation years. As indicated by a pattern correlation coefficient of ~0.99, the orientation and spatial extent of the North Pacific and North Atlantic jets at the 250-hPa level are well replicated by MPAS (Fig. 5a–b). While the North Atlantic jet maximum is slightly stronger in the MPAS simulations, the general strength of both features compares well between the

simulations and reanalysis (Fig. 5a–b). Furthermore, examination of a cross-section of zonally averaged zonal wind shows the jet maximizes at roughly the same altitude and latitude (~200 hPa and ~30º N) in both the MPAS simulations and the ERA-I climatology, albeit the MPAS maximum is moderately weaker (Fig. 5c–d). Additionally, semi-permanent maritime



SLP features, such as the Aleutian Low over the Bering Sea, the North Pacific Subtropical High, and the Icelandic Low, are well captured in the MPAS simulations (Fig. 5a–b). The Bermuda High in the North Atlantic, however, while evident in the MPAS simulations, is comparatively weaker than analysed.

### 3.3 Large-scale Precipitation

Average precipitation over the ten simulation years compared to the 19-year (1998–2016) climatology from the Tropical Rainfall Measurement Mission (TRMM; Huffman et al., 2007) 3B42 product is shown in Figure 6. Because four of our ten simulation years occur before the TRMM record began, we opted to use the full TRMM climatology for comparison. A pattern correlation coefficient of ~0.95 indicates that MPAS simulates the general spatial pattern of tropical precipitation well; the Intertropical Convergence Zone (ITCZ) in the equatorial Pacific and maxima along the west coast of India, over the
Himalayas, and throughout northern South America are all well-represented by the model. The primary difference between the two precipitation fields is the overproduction of precipitation by MPAS in many areas (Fig. 6c), an issue common among other high-resolution modelling studies (e.g., Bacmeister et al., 2014; Small et al., 2014). The overestimation in the subtropical Pacific basin (Fig. 6c) is primarily due to overproductions of summer and fall precipitation. The summer season is also responsible for the overproduction of precipitation through the Bay of Bengal and Gulf of Thailand, suggesting an
overactive summer monsoon in the MPAS simulations. Another notable difference between MPAS and TRMM annual average precipitation is the westward shift of the heaviest precipitation along the ITCZ in the Atlantic basin. This shift in precipitation is likely related to a westward shift of the summertime African Easterly Jet (AEJ; not shown).

### 4 Model Climate: Tropical Cyclones

Tropical cyclones epitomise the high-impact weather phenomena that our simulations are designed to address; therefore, we
consider tropical cyclones as an appropriate exemplar of high-impact weather systems, in order to explore the usefulness of our simulations for examining the effects of climate change on such phenomena. To analyse how TCs appear in our model, we track simulated Northern Hemisphere tropical cyclones using the TempestExtremes objective, feature-based tracking algorithm (Ullrich and Zarzycki, 2017; Zarzycki and Ullrich, 2017). TCs are initially detected as minima in SLP, and then retained as candidate cyclone centres if certain criteria are met. Here, we require that TCs must have a 2 hPa closed SLP
contour within 2º of the storm centre and a 300–500 hPa geopotential thickness maximum within 6º of the storm centre to ensure the presence of a warm core. Additionally, TCs must not travel more than 6º within a 6-h period, must have a lifetime of at least two days, must be located over water for at least 12 h, must have at least two days of 10-m wind speed of at least 14 m/s, and are required to have a genesis latitude south of 45º N. Trajectories that end and begin within 12 h of each other are merged together to prevent broken tracks from being counted twice. Once TC tracks have been obtained, TCs are
separated into basins (Fig. 1) based on their genesis location.



### 4.1 Strength

We compare simulated TC characteristics to the International Best Track Archive for Climate Stewardship (IBTrACS; Knapp et al., 2010). Only the IBTrACS for the ten simulated years (Table 1) are considered for comparison. Consistent with similar studies (e.g., Murakami et al., 2015; Roberts et al., 2015; Yamada et al., 2017), model storms are generally weaker

than observed in terms of maximum 10-m wind speed; several simulated storms do, however, attain a minimum SLP of less than 900-hPa (Fig. 7). Therefore, as in Roberts et al. (2015), storm intensity for the simulated TCs is measured by the minimum lifetime SLP of the storm in addition to maximum 10-m wind speed as defined by the Saffir-Simpson scale. Using the minimum SLP, categories are defined as >994 hPa for tropical storms ($TS_p$) and 980–994 hPa, 965–979 hPa, 945–964 hPa, 920–944 hPa, and <920 hPa for category ($Cat_p$) 1–5 equivalent tropical cyclones, respectively (Roberts et al., 2015).

The subscript p is used to discriminate the SLP-based categories from those defined by the Saffir-Simpson wind speed thresholds. For IBTrACS, the maximum 10-m wind speed and minimum SLP across all reporting centres are used as the observed storm intensity for categorization.

Figure 8 shows the average TC frequency over the ten simulation years for the Northern Hemisphere as a whole, in addition to each basin. Our MPAS simulations generate excess TC activity in the Northern Hemisphere, primarily due to the

over-activity in the Western North Pacific basin. Simulated TC frequencies for the North Atlantic, Eastern North Pacific, and Northern Indian basins are within the observed range. Across all basins, when categorizing TCs by minimum SLP (Fig. 8a), MPAS generally underestimates the number of weak systems (those with strengths less than $Cat_p1$), and overestimates the number of $Cat_p1$ and $Cat_p2$ storms. The frequencies $Cat_p3$ TCs and stronger, on the other hand, are simulated reasonably well. With regard to TC categorization by maximum 10-m wind speed (Fig. 8b), MPAS simulates the frequency of TS

strength TCs quite well in all basins. Strong TCs (Cat4 and Cat5), however, are universally underestimated by the model in favour of Cat1–Cat3 TCs.

### 4.2 Location

Spatially, the model simulated TC track density compares reasonably well with observations; the pattern correlation coefficient is about 0.7 (Fig. 9). The most prominent difference is the lack of TC activity in the eastern portion of the North

Atlantic basin, which is common among several similar modelling studies (e.g., Bell et al., 2013; Strachan et al., 2013; Small et al., 2014; Roberts et al., 2015). TC genesis in this region typically occurs during August and September (Kossin et al., 2010; Daloz et al., 2015); comparison between the simulated atmosphere and ERA-I monthly-averaged 850–200 hPa vertical wind shear for the ten simulation years during these months shows a strong positive bias in the model over the North Atlantic development region that is likely a primary factor in this lack of TC generation (not shown). Figure 9 does not show

a strong track density bias in the Gulf of Mexico. Roberts et al. (2015) note that a steady supply of vorticity in the Caribbean contributed to their overestimation of track density in this area; thus, it is possible that tracking TCs as SLP minima, rather than maxima in 850 hPa relative vorticity helps alleviate this bias. Unlike previous studies (e.g., Small et al., 2014;



Murakami et al., 2015; Roberts et al., 2015; Yamada et al., 2017), we do not find a positive track density bias in the central North Pacific; instead, we see a slight underrepresentation of TC activity in that area around ~150º W.

### 4.3 Seasonal Cycle

Aside from the underestimation in August and September (likely attributed to the lack of TC genesis in the eastern portion of the basin), MPAS simulates the present-day seasonal TC cycle for the North Atlantic reasonably well; TC activity increases during the spring and summer seasons and reaches a maximum in the fall (Fig. 10a). For the Eastern North Pacific, MPAS produces too many storms in the springtime (April and May) and too few storms during the summer months (Fig. 10b). As defined by Camargo et al. (2008), cluster 2 type Eastern Pacific TCs form off the coast of Mexico, travel towards the northwest along the coastline, and have a bimodal seasonal distribution with peaks in late spring/early summer and early fall, similar to the modelled cycle in Fig. 10b. Compared to the ERA-I ten-year climatology, enhanced westerlies at the 500- and 850-hPa levels between 0–20º N in the eastern portion of the Eastern Pacific basin for April and May (not shown) suggest that our simulations may be in a regime more conducive to these cluster 2 storms. For the Western North Pacific (Fig. 10c), MPAS correctly simulates the fall peak in TC activity; there is, however, a secondary peak in April that does not match observations. Although there is a general overestimation of storm activity in the Northern Indian basin, the model does replicate the shape of the seasonal cycle with both the early summer and mid-fall peaks represented, albeit the fall peak occurs one month earlier than observed (Fig. 10d).

### 5 Climate Change Representation

To ensure that our simulations are useful in studying climate change effects on weather phenomena, we compare temperature change fields with large-scale warming patterns generated by a subset of IPCC GCMs. Previous theoretical and modelling studies demonstrate that the Arctic region will continue to warm at a faster rate than the rest of the globe in response to an increase in greenhouse gases (IPCC, 2013, §12.4.2.2). This polar amplification effect is captured in our simulations with portions of the Arctic experiencing temperature changes greater than 16 K compared to differences ≤10 K elsewhere (Fig. 11a). However, we note that Arctic temperatures in our present-day simulations were colder compared to ERA-I, while the future simulations resulted in Arctic temperatures comparable to those produced by GCMs (not shown). As a result, the MPAS simulations produce a larger magnitude of warming in the Arctic compared to the GCM ensemble (Fig. 11a–b). Another result consistent with theory and previous modelling studies is the development of a warming maximum in the tropical upper-troposphere (IPCC, 2013, §12.4.3.2). This area of warming, which occurs between the ~400- and ~150-hPa levels and which maximizes around the 250-hPa level (IPCC, 2013, §12.4.3.2), has been shown to partially mitigate projected increases in TC intensity associated with warming (e.g., Knutson and Tuleya, 1999; Shen et al., 2000; Hill and Lackmann, 2011). As shown in Fig. 11c, this warming signature is replicated in the MPAS simulations.





## 6 Summary and Conclusions

We present a novel set of model simulations produced using MPAS in current and future thermodynamic environments that is designed to maximize our ability to analyse changes in high-impact weather systems; such changes will be reported in future studies. Our use of a global model eliminates the lateral boundary constraints of regional models, while inclusion of

high-resolution, analysed SSTs preserves realistic SST gradients throughout the duration of the simulations. Furthermore, a grid length of 15 km offers an advantage over coarser modelling studies to better represent the atmospheric mesoscale. The future climate simulations employ a technique that combines methods associated with PGW and time-slice experiments; this allows for the inclusion of high-resolution SSTs, plausible future sea ice fields, and seamless simulation of non-consecutive years without excessive spin-up time. While the primary purpose of our simulations is to study climate change effects on

Northern Hemispheric high-impact weather, to achieve this it is necessary to first evaluate the model climate in regard to present-day large-scale circulations as well as the large-scale responses to warming in our climate change experiments; reasonable representation of these aspects is essential to justify moving forward to investigate smaller-scale, high-impact phenomena.

Key results from these simulations include the ability of MPAS to reproduce Northern Hemispheric wintertime

midlatitude storm tracks (Fig. 4) along with semi-permanent maritime SLP and upper-tropospheric jet features (Fig. 5). Tropical characteristics, such as precipitation along the ITCZ in the equatorial Pacific, are also well simulated (r ~ 0.95), although the ITCZ representation in the Atlantic does not compare as favourably to observations (Fig. 6). In regard to TC strength, MPAS is able to produce several tropical cyclones of Cat4 strength, as defined by traditional maximum 10-m wind speed thresholds of the Saffir-Simpson scale (Fig. 7 and Fig. 8b). Categorizing TCs using the minimum SLP thresholds of

Roberts et al. (2015), on the other hand, shows simulated TCs across the full intensity spectrum, including $Cat_p5$ storms (Fig. 7 and Fig. 8a).

While MPAS overestimates TC activity in the Western North Pacific, TC frequency in other Northern Hemispheric basins is within the range of observations (Fig. 8). The largest discrepancy in the simulated spatial distribution of TCs is the lack of TC genesis in the eastern North Atlantic (Fig. 9), likely due to a positive bias in vertical wind shear (not shown).

Otherwise, TC density patterns match observations reasonably well (r ~ 0.7). Additionally, with the exception of the Eastern North Pacific, the seasonal cycles for the Northern Hemispheric basins are well reproduced (Fig. 10). Last, our future simulations replicate two key warming signatures produced by GCMs: Arctic amplification and the warming maximum in the tropical upper-troposphere (Fig. 11).

With our modelling approach, we strive to contribute to the intersection of weather and climate modelling, and aim

to fill a gap between GCMs, which are unable to simulate small-scale weather phenomena, and high-resolution limited area models, which are constrained by lateral boundaries, to provide the possibility of studying high-impact events in a consistent global context. We anticipate these simulations, in conjunction with similar efforts, will have great value in projecting and understanding changes in high-impact weather phenomena for which dynamics on sub-synoptic scales are important.



Beyond the tropical cyclones described here, this could include flooding rains and damaging winds associated with extratropical cyclones, flooding monsoon rains, and localized droughts and heat waves. Research involving these simulations is currently underway investigating climate change effects on the following phenomena:

- Extratropical transition of TCs
- TC seasonality
- Midlatitude precipitation extremes and windstorms embedded in extratropical cyclones
- Persistent anomalies and blocking

Many more aspects of these simulations, however, remain to be explored. Therefore, we make the simulation output available to the research community as detailed in the Acknowledgements section in the hope that it will be useful to the
broader scientific community for studying various meteorological phenomena, as well as for conducting model comparison studies.

**Data and code availability**

The source code for the model used in this study, MPAS, is freely available from https://mpas-dev.github.io. The MPAS model output from the simulations presented in this manuscript is located on the Cheyenne High Performance Storage
System (HPSS) and on the NCSU Henry2 Cluster. Please contact the corresponding author for additional details on accessing this data. The TempestExtremes tracking algorithm used in this study is available from https://github.com/ClimateGlobalChange/tempestextremes. ECMWF Interim Reanalysis can be obtained from https://rda.ucar.edu/datasets/ds627.0. TRMM 3B42 data can be accessed from http://disc.sci.gsfc.nasa.gov/TRMM. PRISM data is available from http://prism.oregonstate.edu. IBTrACS version v03r10 can be obtained from
https://www.ncdc.noaa.gov/ibtracs.

**Author contributions**

All authors contributed equally in designing the model experiments. A. Michaelis conducted and analysed the simulations. A. Michaelis wrote the manuscript with editorial modifications from G. Lackmann and W. Robinson.

**Competing interests**

The authors declare they have no conflicts of interest.



## Acknowledgements

This research was supported by NSF grants AGS-1546743 and AGS-1560844, awarded to North Carolina State University (NCSU). The MPAS and NCAR Command Language (NCL) were made available by the National Center for Atmospheric Research (NCAR), sponsored by the National Science Foundation (NSF). High-performance computing support from

Cheyenne (doi:10.5065/D6RX99HX) was provided by NCAR's Computational and Information System Laboratory, sponsored by the NSF. Our custom MPAS grid and additional MPAS support was provided by Michael Duda at NCAR. The CMIP5 GCM ensemble mean data and interpolation codes used in this study were provided by Chunyong Jung at NCSU. Model output from the simulations presented in the manuscript is located on the Cheyenne High Performance Storage System (HPSS) and on the NCSU Henry2 Cluster. Please contact the corresponding author for additional details on

accessing this data.

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



**Table 1: Average Multivariate ENSO Index (MEI), Oceanic Niño Index (ONI), and Corresponding ENSO Phase During the TC Season (June–November) for the Chosen Simulation Years.**

| Year | Multivariate ENSO Index (MEI) Rank: JJ–ON average | Oceanic Niño Index (ONI): JJA–SON average | ENSO phase |
|---|---|---|---|
| 2010 | 3.8 | -1.2 | Strong La Niña |
| 1988 | 6.6 | -1.2 | Strong La Niña |
| 2011 | 16.2 | -0.7 | Weak La Niña |
| 2013 | 26.8 | -0.2 | Neutral |
| 2001 | 31.8 | -0.1 | Neutral |
| 2005 | 34.2 | 0.0 | Neutral |
| 1992 | 47.5 | 0.3 | Neutral |
| 1994 | 57.1 | 0.5 | Weak El Niño |
| 2015 | 64.8 | 1.7 | Strong El Niño |
| 1997 | 66.0 | 1.8 | Strong El Niño |

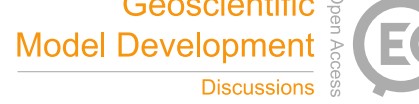


**Table 2: List of 20 CMIP5 GCMs Used to Compute Ensemble Mean Temperature "Deltas" and Sea Ice Fields.**

| Model | Modeling Center/Group | Grid Length |
|---|---|---|
| ACCESS1-0 | Commonwealth Scientific and Industrial Research Organization (CSIRO) and Bureau of Meteorology (BOM), Australia | 1.25º x 1.875º |
| ACCESS1-3 | | |
| CanESM2 | Canadian Centre for Climate Modeling and Analysis | 2.8º x 2.8º |
| CMCC-CM | Centro Euro-Mediterraneo sui Cambiamenti Climatici (Euro-Mediterranean Center on Climate Change) | 0.8º x 0.8º |
| CNRM-CM5 | National Centre of Meteorological Research, France | 1.4º x 1.4º |
| GISS-E2-H | | |
| GISS-E2-H-CC | NASA Goddard Institute for Space Studies | 2º x 2.5º |
| GISS-E2-R-CC | | |
| HadGEM2-AO | Met Office Hadley Centre | 1.25º x 1.875º |
| HadGEM2-ES | | |
| INMCM4 | Institute for Numerical Mathematics | 1.5º x 2.0º |
| IPSL-CM5A-MR | Institut Pierre Simon Laplace, France | 1.25º x 2.5º |
| IPSL-CM5B-LR | | 1.8º x 2.75º |
| MIROC-ESM | Japan Agency for Marine-Earth Science and Technology, Atmosphere and Ocean Research Institute (The University of Tokyo), and National Institute for Environmental Studies | 2.8º x 2.8º |
| MIROC-ESM-CHEM | | |
| MPI-ESM-LR | Max Planck Institute for Meteorology | 1.8º x 1.8º |
| MPI-ESM-MR | | |
| MRI-ESM1 | Meteorological Research Institute, Japan | 1.1º x 1.1º |
| NorESM1-M | Norwegian Climate Center, Norway | 1.9º x 2.5º |
| NorESM1-ME | | |



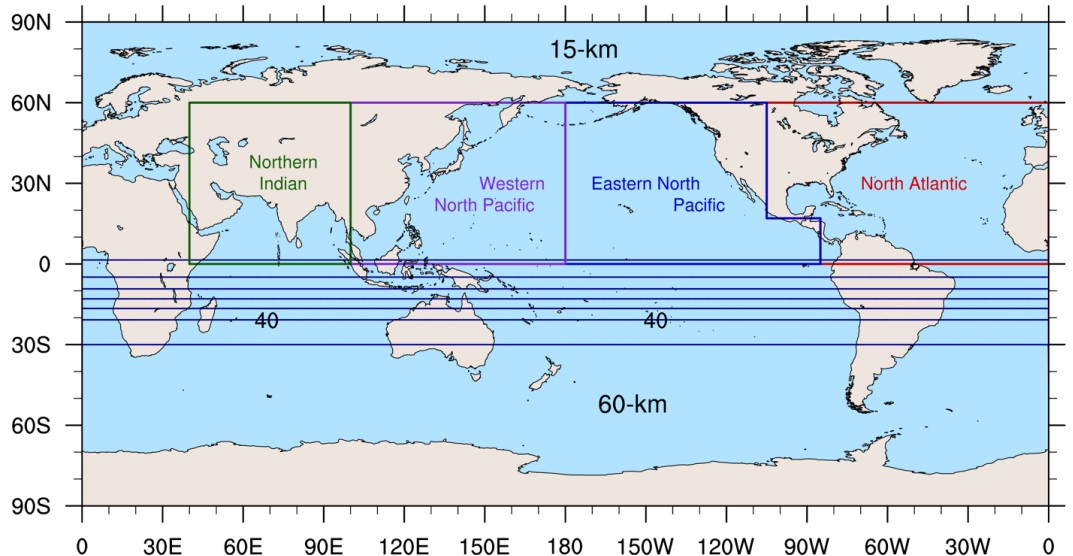

**Figure 1: Variable-resolution mesh for MPAS simulations and geographical regions of the tropical cyclone basins defined in this study.**



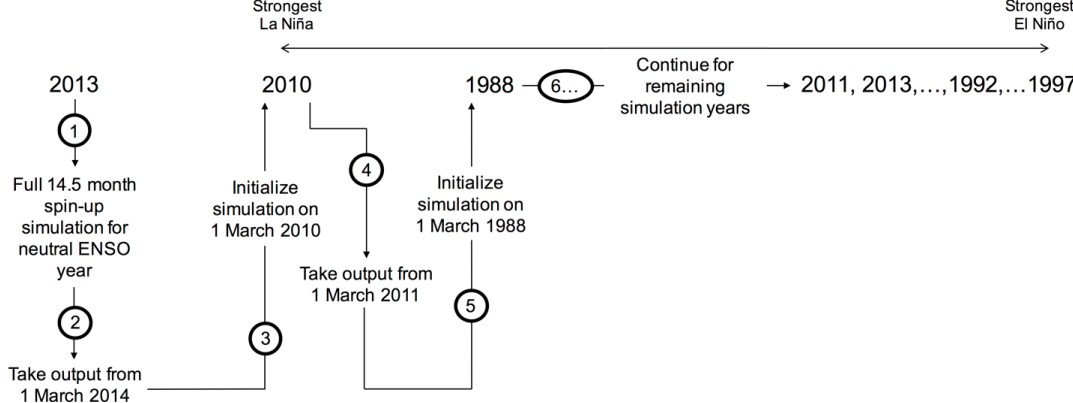

Figure 2: Flow chart depicting the "daisy-chain" simulation method.



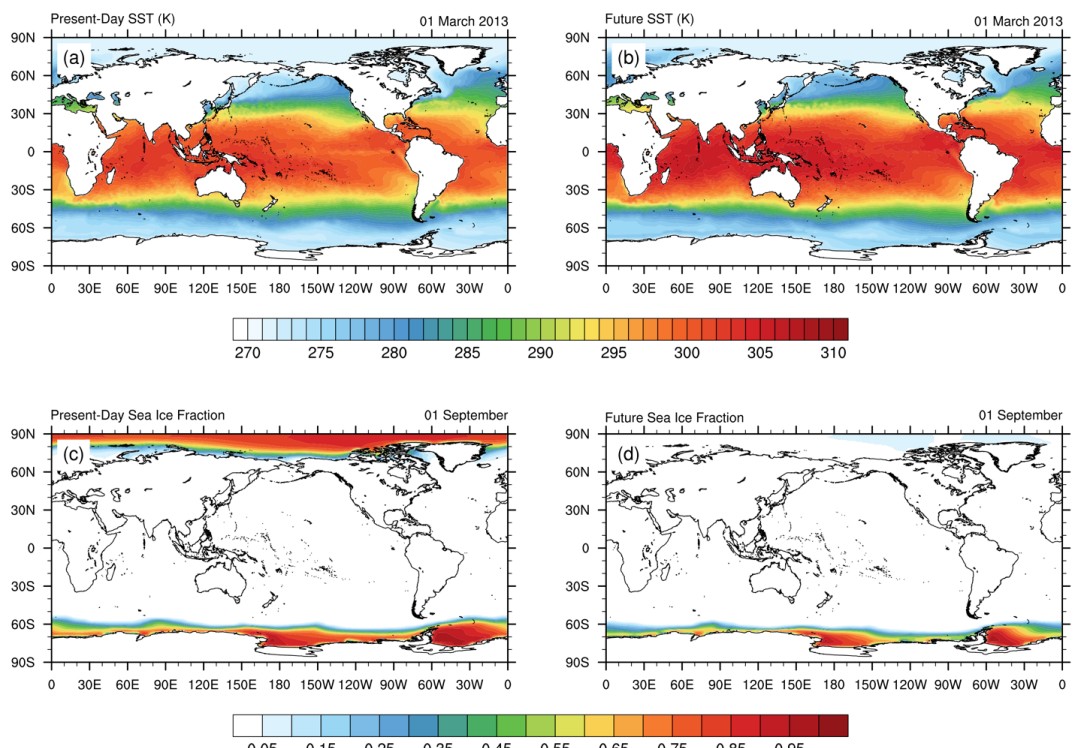

**Figure 3: Example SST (K) on 1 March 2013 for (a) present-day and (b) future MPAS simulations and example sea ice fraction on 1 September for the (c) present-day and (d) future MPAS simulations. Contours are shaded every 1 K in (a) and (b) and every 0.05 units in (c) and (d).**



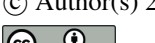

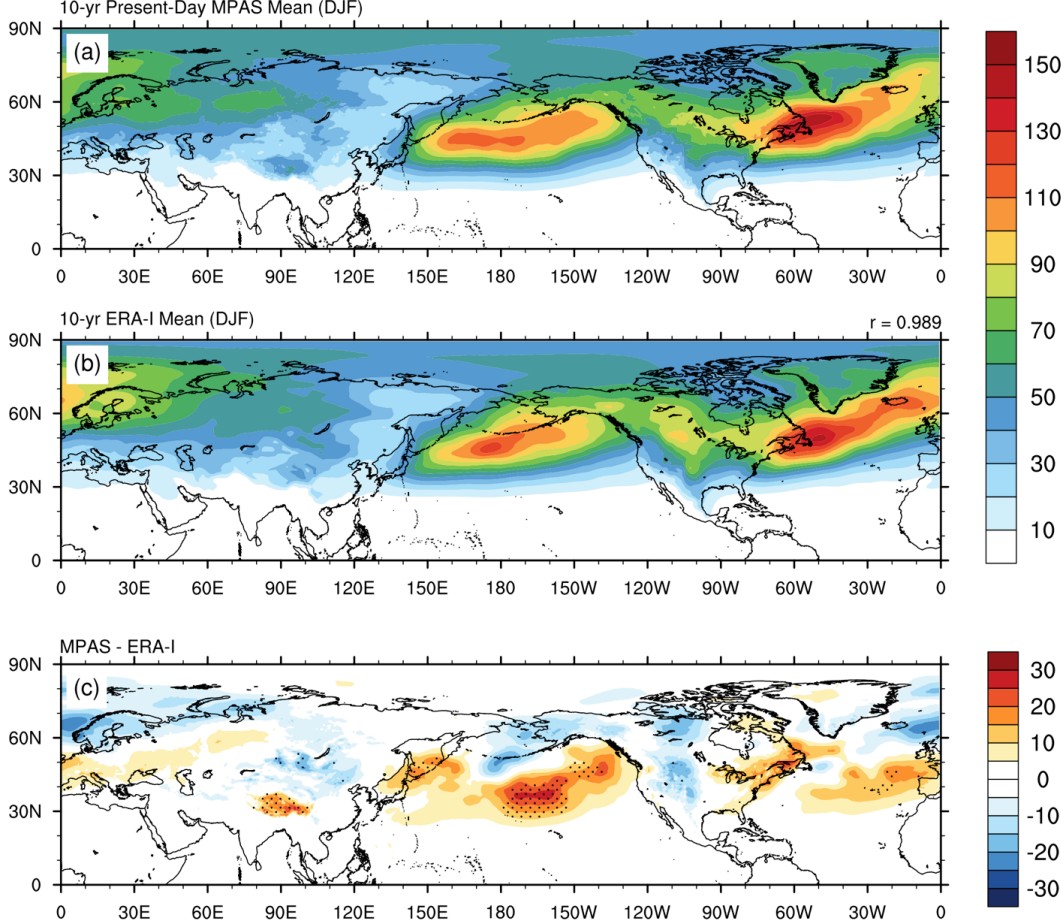

**Figure 4:** Average wintertime (DJF) SLP variance (hPa$^2$) over the ten simulation years for (a) present-day MPAS simulations, (b) ERA-I 10-yr climatology, and (c) the model bias (MPAS minus ERA-I). Contours are shaded every 10 hPa$^2$ in (a) and (b) and every 5 hPa$^2$ in (c). MPAS output were linearly interpolated to the ERA-I grid for point-to-point comparison. The pattern correlation coefficient is reported at the top-right of (b). Stippling in (c) indicates locations where the MPAS 10-yr mean exceeds the range computed from 100 random samples of 10-yr means from ERA-I by more than 5% (~7 hPa$^2$).





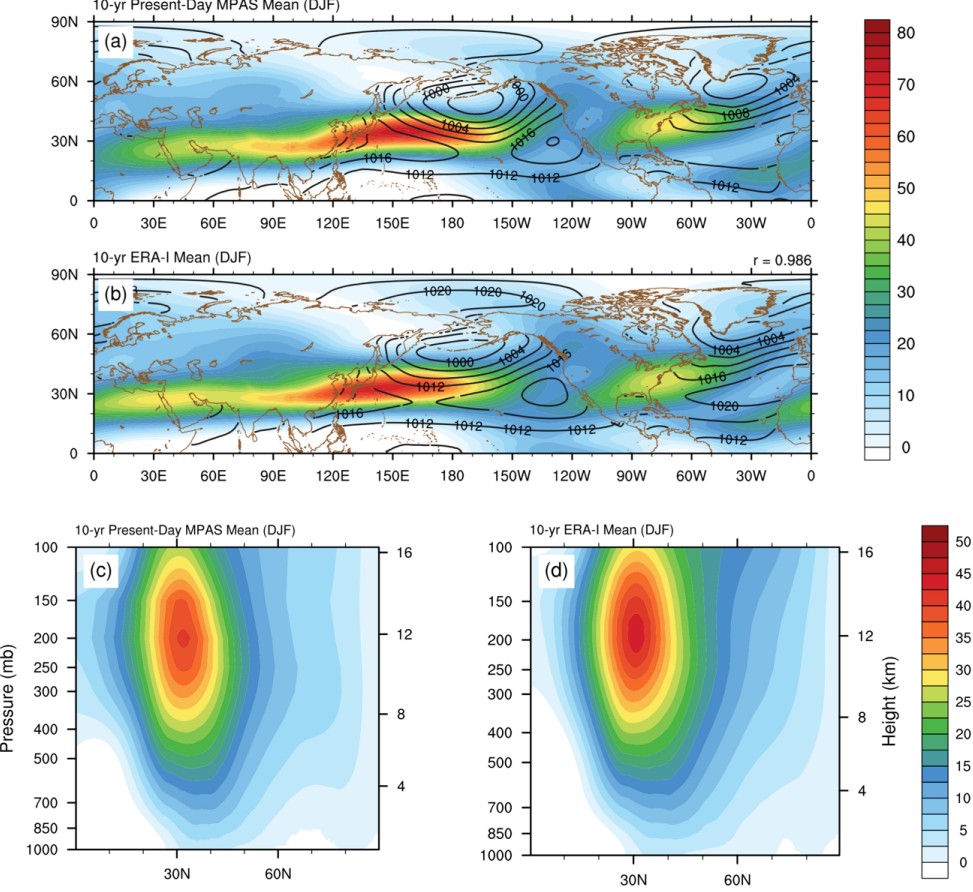

**Figure 5: (Top row) Average wintertime (DJF) 250-hPa zonal wind speed (ms⁻¹; shaded every 2.5 ms⁻¹) and SLP (hPa; contoured every 4 hPa) over the ten simulation years for (a) present-day MPAS simulations and (b) ERA-I 10-yr climatology. (Bottom row) Average wintertime (DJF) cross-section of zonally averaged zonal wind speed (ms⁻¹; shaded every 2.5 ms⁻¹) for (c) present-day MPAS simulations and (d) ERA-I 10-yr climatology. The pattern correlation coefficient for 250-hPa zonal wind speed is reported at the top-right of (b). SLP contours in (a) and (b) are masked over land owing to noise in areas of complex terrain.**



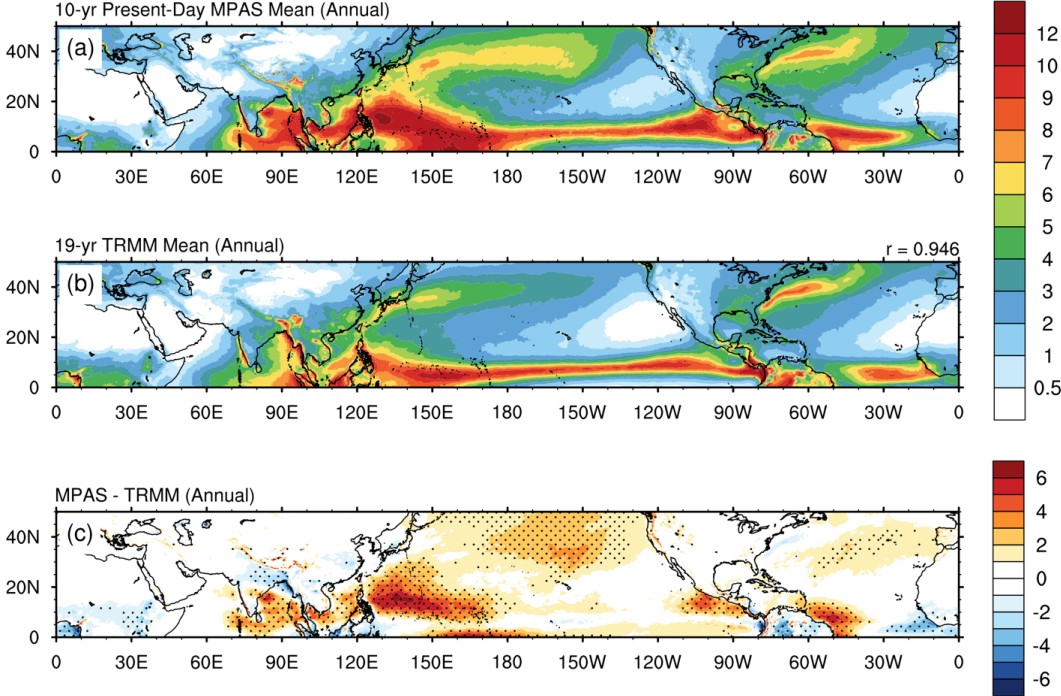

**Figure 6: Average total annual precipitation (mm day$^{-1}$) for (a) present-day MPAS simulations, (b) TRMM 19-yr climatology, and (c) the model bias (MPAS minus TRMM). MPAS output were linearly interpolated to the TRMM grid for point-to-point comparison. Pattern correlation coefficient is reported in the top-right of (b). Stippling in (c) indicates locations where the MPAS 10-yr mean exceeds the range computed from 100 random samples of 10-yr means from TRMM by more than 5% (~1.2 mm day$^{-1}$).**





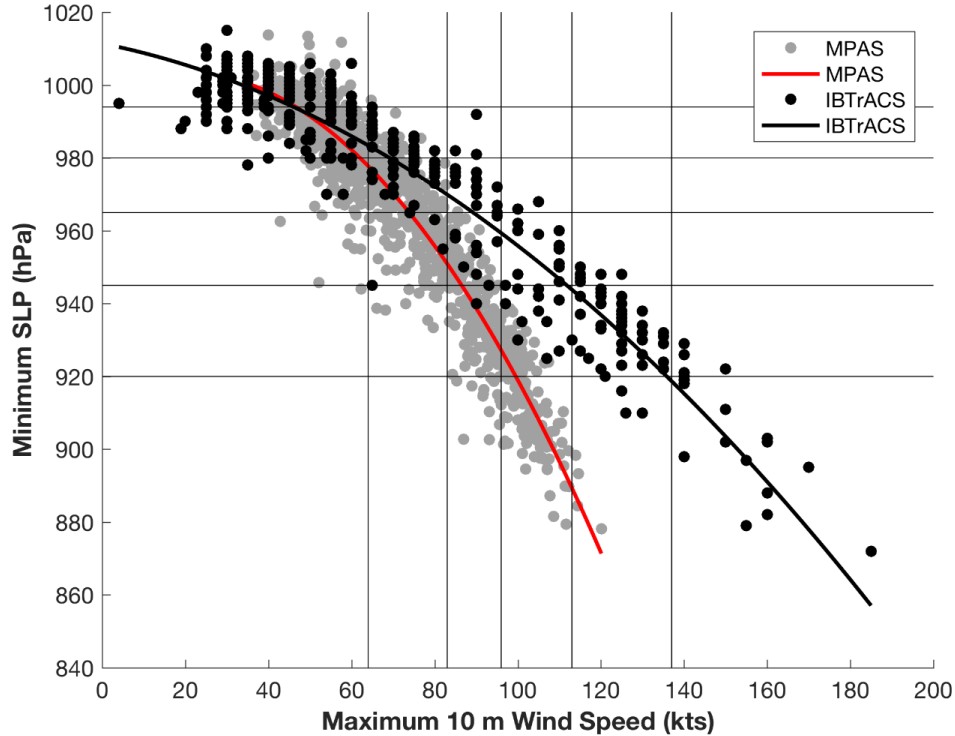

**Figure 7: Scatter plot of maximum 10-m wind speed (kts) versus minimum SLP (hPa) for IBTrACS (black) and present-day MPAS (grey) Northern Hemispheric TCs. The lines of best fit for each (IBTrACS in black and MPAS in red) were computed using a second-order polynomial. The wind (SLP) category thresholds are indicated by the vertical (horizontal) lines.**



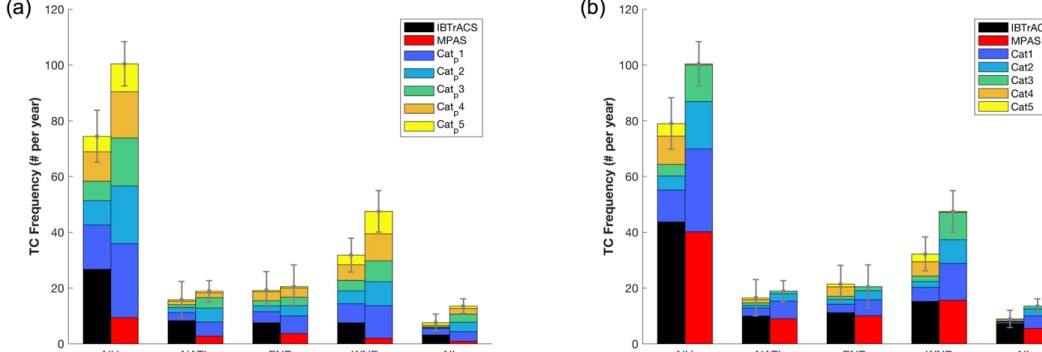

**Figure 8: Average number of TCs over the ten simulation years for the Northern Hemisphere and each Northern Hemisphere basin. Columns are coloured by intensity categories based on (a) minimum lifetime SLP and (b) maximum lifetime 10-m wind speed. The bottom colour represents intensities of tropical storm strength or less for IBTrACS and MPAS in the first and second columns, respectively. Categories 1–5 are shaded for both data sets according to the legend. The error bars indicate the interannual standard deviation. The number of TCs for IBTrACS varies based on strength metric due to the lack of SLP records for a select number of storms.**



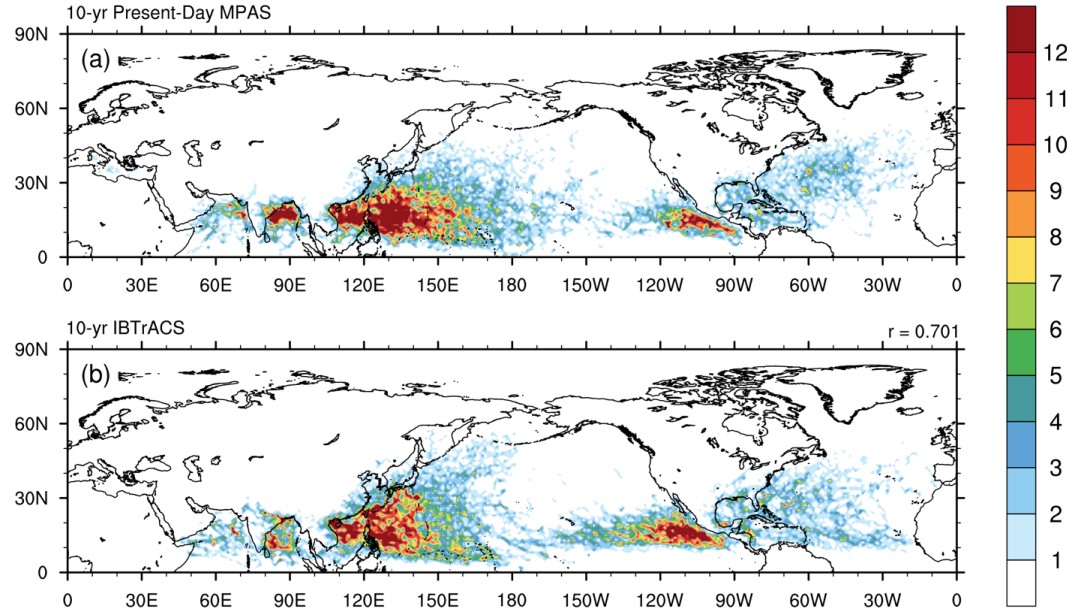

**Figure 9: Track density (number of cyclone tracks per 1° x 1° area) over the ten simulated years for (a) present-day MPAS simulations and (b) IBTrACS. Contours are shaded every 1 count.**





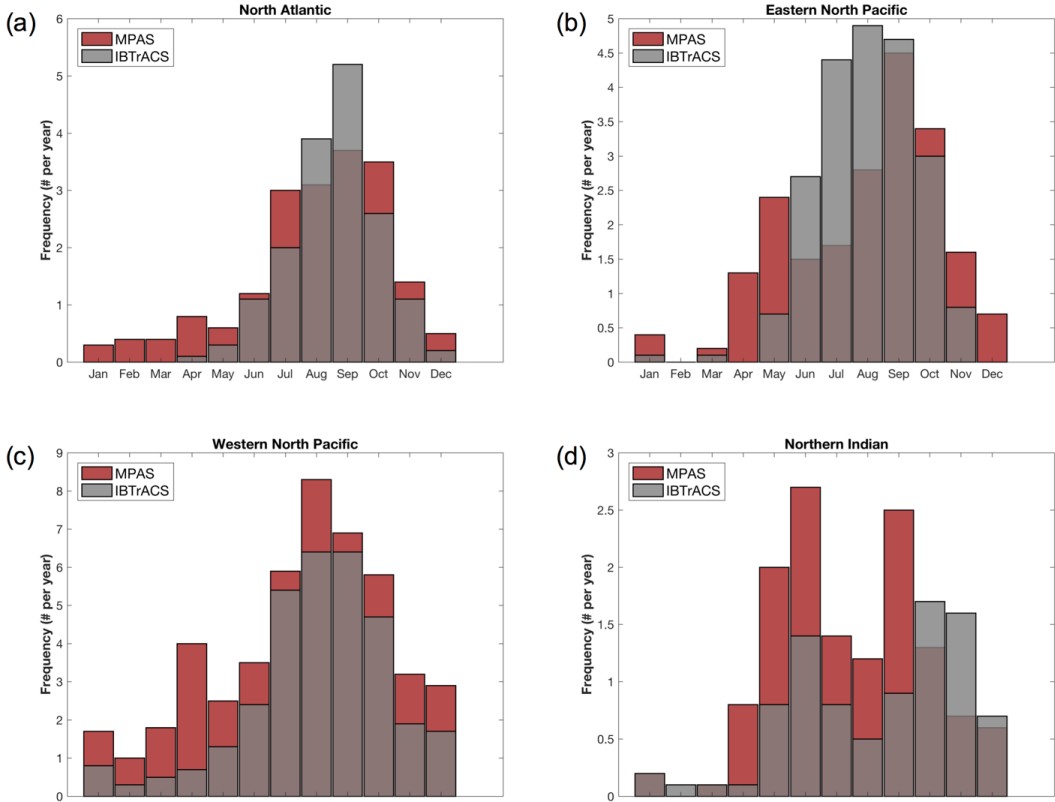

**Figure 10: Monthly average TC frequency over the ten simulated years for the (a) North Atlantic, (b) Eastern North Pacific, (c) Western North Pacific, and (d) Northern Indian basins. The frequencies for IBTrACS (MPAS simulations) are shown in the grey (red) bars.**





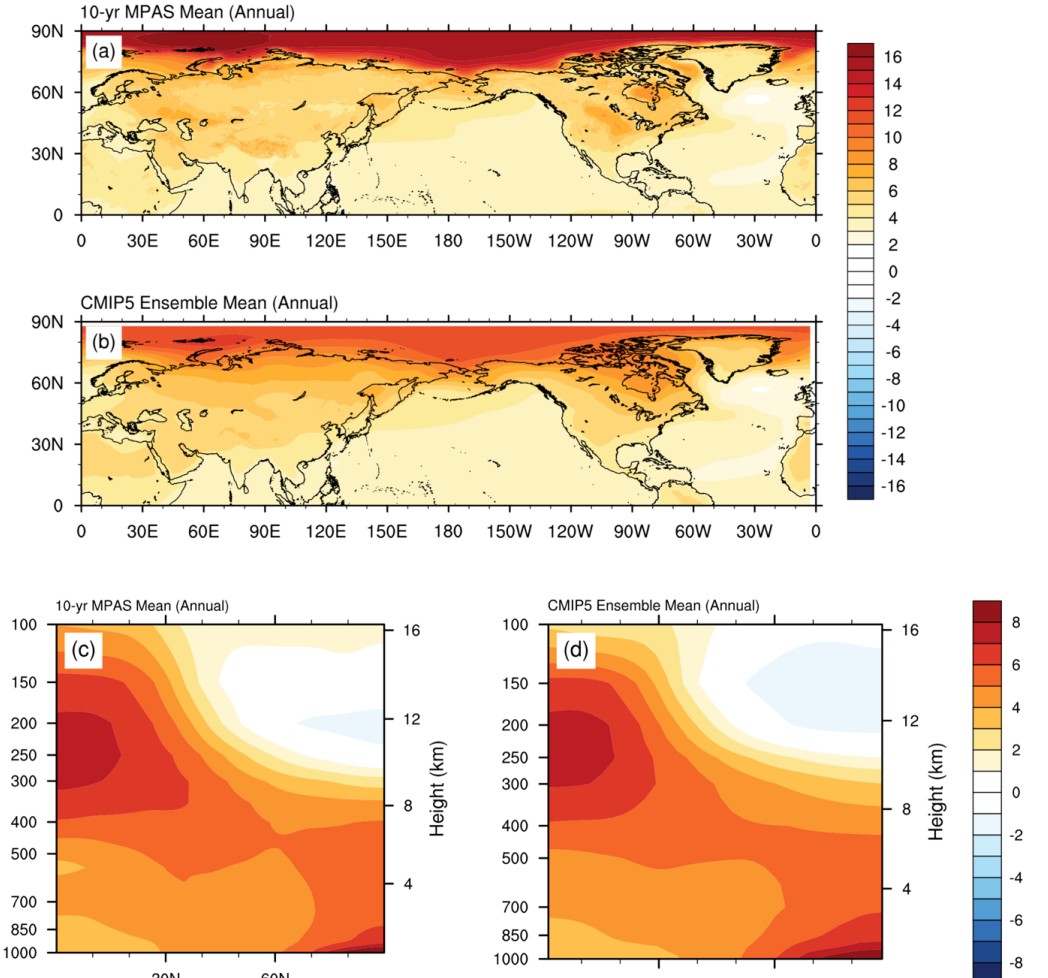

**Figure 11: (Top row)** Average annual 2 m temperature difference (K; future minus current) for the (a) MPAS simulations and (b) CMIP5 GCM ensemble mean. **(Bottom row)** Average annual difference cross-section of zonal mean temperature (K; future minus current) for the (c) MPAS simulations and (d) CMIP5 GCM ensemble mean. Contours are shaded every 1 K in all panels.