# Peer review of "Evaluation of a Unique Approach to High-Resolution Climate Modelling using the Model for Prediction Across Scales-Atmosphere (MPAS-A) version 5.1"

_Geoscientific Model Development, 2019_

## Short Comment (SC1) · 3 Apr 2019

Dear Drs. Michaelis, Lackmann and Robinson,

A very interesting paper. As a member of the group at Los Alamos National Lab that develops 3 other MPAS components—the ocean (MPAS-O), sea ice (MPAS-SI) and land-ice (MALI) models—it is always nice to see an analysis of MPAS components.

MPAS itself is a framework for developing geophysical models rather than a model itself. I would therefore recommend changing the title to either: Evaluation of a Unique Approach to High-Resolution Climate Modelling using the Model for Predic-
tion Across Scales (MPAS) Atmosphere version 5.1 or Evaluation of a Unique Approach to High-Resolution Climate Modelling using the Model for Prediction Across Scales Atmosphere (MPAS-A) version 5.1 Similarly, in the abstract, "Model for Prediction Across Scales-Atmosphere (MPAS)" should be "Model for Prediction Across Scales-Atmosphere (MPAS-A)" and throughout the text "MPAS" should be "MPAS-A"

Thanks very much for your consideration. Xylar

---

## Short Comment (SC2) · 3 Apr 2019

Dear Dr. Asay-Davis,

Thank you for your comments and pointing out this clarification. We will be sure to change instances of "MPAS" to "MPAS-A" in a revised version of the manuscript.

Thanks again, Allison

---

## Referee Comment (RC1) · James Done (Referee) · 6 May 2019

General Comments: This study presents a novel experimental design to advance our understanding of climate change effects on high-impact mid-latitude and tropical weather events. Rather than simulating climates using continuous simulations over many years, the authors selectively sample years of varying ENSO phase, and run under current climate and future climate. Current climate is simply analyzed SSTs and future climate is created by adding a climate change perturbation to the analyzed SSTs, enhancing greenhouse gas concentrations and reducing sea-ice. This Pseudo-Global Warming (PGW) approach has been done many times for regional domains where the

daily synoptic-scale variability is constrained to replicate current climate variability. In this global model context, however, there is no such constraint on the synoptic-scale variability, and the atmospheric variability is free to respond to the future climate.

The authors make a strong case that well-thought-out experimental designs such as this one promise to make significant advances in our understanding of climate change effects on high-impact global weather.

The paper is a pleasure to read – it is well structured, well written, and the figures are clear and useful. The introduction includes a comprehensive review of previously published work, and provides good motivation for the study. The methodology appears sound, and assumptions are acknowledged. The paper contains an important evaluation of the simulated current climate by MPAS – a relatively new global atmospheric model. MPAS is shown to be credible at reproducing many aspects of the observed large-scale dynamics in current climate. I have a few specific issues described below that should be addressed prior to consideration for publication.

Specific Comments: 1) The claim is made that the experiment assesses future thermo-dynamic environments. I understand that the frequency distribution of ENSO phases is the same in current and future climate. But MPAS will permit some large-scale circulation change in response to the future SSTs so I'm not sure circulation change is small compared to the thermodynamic change. Please clarify the contributions of thermodynamic change and circulation change permitted by your experimental setup. Perhaps checking the magnitude of circulation difference fields would help. More generally, can you provide more guidance on how the future change results should be interpreted? How should we interpret future changes based on fixed ENSO phase frequencies but variable atmospheric circulation response?

2) A related issue is that the experimental approach assumes that SSTs in all phases of ENSO change by the same magnitude and spatial pattern. Can you comment on how realistic this assumption is and any implications for the results?

3) Please explain why this study chose to sample ENSO phases and not phases of some other interannual of decadal mode of climate variability?

4) The discussion of missing cool-wakes in the simulations on page 8 is well stated. But that is only half the story. The reanalysis SSTs will contain the cool wakes of observed TCs, so this could unphysically dampen simulated TCs that cross these 'phantom' cool wakes.

5) The evaluation shows MPAS does a reasonable job at capturing the climatological spatial distribution of TCs. Given that your approach emphasizes ENSO phases, can you also check whether MPAS captures the observed TC response to ENSO phase in the Atlantic and Pacific.

6) MPAS misses TCs that develop from easterly waves over the eastern North Atlantic. I agree that one possible reason is enhanced shear. Another reason could be that the westward shift in the African Easterly Jet means that the wave energy accumulation zone (as discussed in Done et al. 2011) is also shifted westward.

Done, J.M., Holland, G.J. and Webster, P.J., 2011. The role of wave energy accumulation in tropical cyclogenesis over the tropical North Atlantic. Climate dynamics, 36(3-4), pp.753-767.

7) The purpose of Fig. 10 is to compare seasonal cycles of TCs. I suggest plotting the normalized distributions to remove the effect of differences in absolute numbers.

8) The motivation for this experimental design is to assess future changes in high-impact weather. But no results on future changes to high impact weather are presented. I read in the final section that results will be forthcoming in a separate publication. To set readers expectations I suggest adding a note earlier in the manuscript, perhaps at the end of the introduction.

Technical corrections: None

---

## Referee Comment (RC2) · Anonymous Referee #2 · 15 May 2019

Review "Evaluation of a unique approach to high-resolution climate modelling using the model for prediction across scales (MPAS) version 5.1" by Michaelis et al.

This manuscript describes a set of simulations and their basic results with the global high-resolution model MPAS 5.1. The set-up is of the experiments is interesting and in some aspects new. The results presented are sound, well represented and illustrate the very good quality of the model. I have however a few comments.

1. The authors show the TC characteristics for the present climate, but not for the future climate, whereas they show in section 5 plots of the future climate. I was disappointed that nothing was said about TCs in the future climate. I assume that it will be analysed

in a future paper, but for the reader it feels very disappointing. Why not postpone the climate change simulations altogether to a following paper? The added value of section 5 is very minor. Concentrating on the present climate and showing the quality of the model as in sections 3 and 4 should be fine for presenting the model and the experimental set-up. If you want to discuss climate change I would like it to be more than what is in the paper and include TC changes, or make it explicitly that they will discussed in a a future paper. 2. The experimental set-up is new in certain aspects, but I had the feeling that the authors over state a bit the uniqueness. They present it as global PGW simulations, whereas to me they are a clever way of time slice experiments with prescribed SSTs, with a spin-up to let the model come into equilibrium with the SST and CO2 concentrations. I found the PGW term therefore confusing and suggest to remove it. 3. Ordering the simulations by El-Nino strength is clever, but anomalies outside El-Nino can occur, which can be substantial, and still have to be adjusted. I am not sure if one month spin-up is sufficient for these regions. 4. The use of high resolution SST data and computation of future SST by taking the delta of an ensemble of low resolution GCMs is a good approach. However, the delta SST can be influenced by the biases in SST. This can be particular large at the western boundary currents, that are incorrectly represented by low-resolution GCMs.

---

## Short Comment (SC3) · 15 May 2019

I am writing as an executive editor of GMD to highlight several issues with the code availability section which needs to be remedied in the revised manuscript.

Configuration files, run scripts, and analysis scripts are missing

The code and data availability section refers to MPAS, and to various data sets. However this is insufficient to allow the reader to reproduce the results in the manuscript. For this, the reader needs the configuration files or run scripts used to run each experiment, and any scripts which were used to post process or analyse the model data. The exact version of these should be persistently and publicly archived (for example on Zenodo) and cited from the manuscript.

Code is on GitHub

The reference to the MPAS code used is on GitHub. This is both impermanent, as MPAS might move off GitHub in the future, and fails to identify the exact version of the code which was used (the model text refers to version 5.1, but was this the 5.1 release, or just a version of the master branch taken when 5.1 was the release number?) To remedy this, the exact version of MPAS used should be publicly and persistently archived. Since MPAS is developed on GitHub, the GitHub-Zenodo integration may be the easiest way to accomplish this. See: https://guides.github.com/activities/citable-code/.

Similar issues apply to Tempest Extremes.

Result data is insufficiently identified

The result data is only available "on request". If at all possible, this data should be persistently and publicly archived so that the reader who wishes to investigate a result in the paper can do so directly. However, if this is not possible for licence reasons or because of the data volume, the data needs to be sufficiently precisely identified that it will still be retrievable if the authors have moved on from the host institution.

**0.1 Input data is insufficiently identified and incorrectly cited**

The external data sets used are incorrectly cited, and in some cases poorly identified. Specifically:

1. ERA-Interim data. The NCAR data archive provides DOIs for this data, and even tells you what to write in the bibliography. Please cite this in accordance with their instructions.

2. TRMM data. Similarly, there is a data citation tab which tells you how to cite this data correctly.

3. PRISM appear not to have precise data citation instructions, however they do have a precise convention for identifying the exact data set(s) used.

General form of citations

Citations for code and data should be in the form of references in the manuscript bibliography, which are cited from the code and data availability section. See the best practice section in: https://www.geoscientific-model-development.net/about/code_and_data_policy.html

---

## Author Comment (AC1) · 24 Jun 2019

**Response to Reviewer 1 (James Done)**

We'd like thank Dr. Done for his insightful review of our manuscript, including his positive comments and suggestions for improvement. Responses to his suggestions are included below.

1. The claim is made that the experiment assesses future thermodynamic environments. I understand that the frequency distribution of ENSO phases is the same in current and future climate. But MPAS will permit some large-scale circulation change in response to the future SSTs so I'm not sure circulation change is small compared to the thermodynamic change. Please clarify the contributions of thermodynamic change and circulation change permitted by your experimental setup. Perhaps checking the magnitude of circulation difference fields would help. More generally, can you provide more guidance on how the future change results should be interpreted? How should we interpret future changes based on fixed ENSO phase frequencies but variable atmospheric circulation response?

It is true that we are assessing more than just thermodynamic changes since MPAS is a global model allowing for circulation changes to occur. We have removed "thermodynamic" from L8 on page 1, L11 on page 2, L3 and L29 on page 6, L9 and L34 on page 8, and L2 on page 13. We have also added L12–15 on page 2 to further clarify our methods.

In regard to the circulation changes, please see Figs. 1 and 2 below. Fig. 1 shows a stronger wintertime Pacific jet and a deeper Aleutian Low in the future simulations. Changes in the Atlantic are less severe, but show a weaker jet over the northeast US and a slightly weaker Icelandic Low. The cross-sections in Fig. 2 show an overall stronger jet in the future simulations, with a poleward shift in the maximum.

Given GCM difficulties in representing ENSO, we felt that the approach taken was the most defensible. Future research will need to determine how ENSO phase frequencies will change with warming, but we agree that this is another assumption in our methods and have added a sentence acknowledging this on page 7, L27.

Figure 1. Average wintertime (DJF) 250-hPa zonal wind speed ( $ms^{-1}$ ; shaded) and SLP (hPa; contoured) over the ten MPAS simulation years for (a) present-day, (b) future, and (c) future minus present-day 250-hPa wind speed (shaded) and SLP (contours). Wind contours are shaded every 2.5  $ms^{-1}$  in (a), (b) and every 2  $ms^{-1}$  in (c). SLP is contoured every 4 hPa in (a), (b) and every 2 hPa in (c).